# Intraoperative Contrast-Enhanced Ultrasonography (Io-CEUS) in Minimally Invasive Thoracic Surgery for Characterization of Pulmonary Tumours: A Clinical Feasibility Study

**DOI:** 10.3390/cancers15153854

**Published:** 2023-07-29

**Authors:** Martin Ignaz Schauer, Ernst-Michael Jung, Natascha Platz Batista da Silva, Michael Akers, Elena Loch, Till Markowiak, Tomas Piler, Christopher Larisch, Reiner Neu, Christian Stroszczynski, Hans-Stefan Hofmann, Michael Ried

**Affiliations:** 1Department of Thoracic Surgery, University Medical Center Regensburg, Franz-Josef-Strauss-Allee 11, 93053 Regensburg, Germanytomas.piler@ukr.de (T.P.); christopher.larisch@ukr.de (C.L.); hans-stefan.hofmann@ukr.de (H.-S.H.); michael.ried@ukr.de (M.R.); 2Institute for Radiology, University Medical Center Regensburg, Franz-Josef-Strauss-Allee 11, 93053 Regensburg, Germany; natascha.platz-batista-da-silva@ukr.de (N.P.B.d.S.); michael.akers@ukr.de (M.A.); christian.stroszczynski@ukr.de (C.S.)

**Keywords:** Io-CEUS, contrast-enhanced ultrasound, intraoperative ultrasound, thoracic surgery, pulmonary tumour

## Abstract

**Simple Summary:**

Finding solitary pulmonary nodules (SPNs) during thoracic surgery, especially during minimally invasive procedures, remains a major challenge. Moreover, in cases of unclear focal findings, the frozen section result must be waited for, which influences the surgical procedure. Therefore, we are investigating for the first time the use of intraoperative contrast-enhanced ultrasound (Io-CEUS) in minimally invasive thoracic surgery to, on the one hand, visualize unclear SPNs, and on the other hand, to characterize the SPNs directly before surgical resection. In the future, Io-CEUS could make “live histology” possible in thoracic surgery, which may influence the surgeon’s intraoperative decisions and the extent of lung resection.

**Abstract:**

Background: The intraoperative detection of solitary pulmonary nodules (SPNs) continues to be a major challenge, especially in minimally invasive video-assisted thoracic surgery (VATS). The location, size, and intraoperative frozen section result of SPNs are decisive regarding the extent of lung resection. This feasibility study investigates the technical applicability of intraoperative contrast-enhanced ultrasonography (Io-CEUS) in minimally invasive thoracic surgery. Methods: In this prospective, monocentric clinical feasibility study, n = 30 patients who underwent Io-CEUS during elective minimally invasive lung resection for SPNs between October 2021 and February 2023. The primary endpoint was the technical feasibility of Io-CEUS during VATS. Secondary endpoints were defined as the detection and characterization of SPNs. Results: In all patients (female, n = 13; mean age, 63 ± 8.6 years) Io-CEUS could be performed without problems during VATS. All SPNs were detected by Io-CEUS (100%). SPNs had a mean size of 2.2 cm (0.5–4.5 cm) and a mean distance to the lung surface of 2.0 cm (0–6.4 cm). B-mode, colour-coded Doppler sonography, and contrast-enhanced ultrasound were used to characterize all tumours intraoperatively. Significant differences were found, especially in vascularization as well as in contrast agent behaviour, depending on the tumour entity. After successful lung resection, a pathologic examination confirmed the presence of lung carcinomas (n = 17), lung metastases (n = 10), and benign lung tumours (n = 3). Conclusions: The technical feasibility of Io-CEUS was confirmed in VATS before resection regarding the detection of suspicious SPNs. In particular, the use of Doppler sonography and contrast agent kinetics revealed intraoperative specific aspects depending on the tumour entity. Further studies on Io-CEUS and the application of an endoscopic probe for VATS will follow.

## 1. Introduction

The intraoperative detection and diagnostic or therapeutically appropriate resection of unclear solitary pulmonary nodules (SPNs) continues to be a major challenge in thoracic surgery, especially in minimally invasive video-assisted thoracic surgery (VATS) [1]. In addition, due to the continuing high incidence of lung cancer and the increased prevalence of lung cancer screening over the past several years, a further increase in SPNs of an unclear entity, which must be histologically confirmed, is to be expected [2]. Consequently, in addition to preoperative staging, we also need reliable and affordable procedures to find SPNs intraoperatively on the non-ventilated, deflated lung and, at best, to characterize them with regard to the entity present. The extent of resection in cases of SPNs of an unclear entity is decisive and continues to depend on the intraoperative frozen section result. The use of intrathoracic ultrasonography during thoracic surgery has been described as a safe method for localizing SPNs in several studies [3,4,5,6,7]. These studies have demonstrated that intrathoracic ultrasound can reliably detect SPNs and showed no disadvantage compared to manual palpation [4,8]. However, there are no data to date on the use of intraoperative contrast-enhanced ultrasound (Io-CEUS) in thoracic surgery, especially in minimally invasive thoracic surgery using VATS. With Io-CEUS, it is hoped that the additional visualization and characterization of the SPNs in deflated lungs based on contrast agent kinetics directly before surgical resection can be achieved [9].

In general surgery, Io-CEUS is already well established in the intraoperative assessment and characterization of hepatic masses and enables the surgeon to maintain optimal resection margins depending on the tumour entity and spread [10]. Here, malignant and benign lesions can be differentiated in the intraoperative setting, which significantly influences the surgical procedure [11,12]. This has already found its way into the international guidelines EFSMUB and WFUMB [13]. Io-CEUS is also successfully used in other specialties [9]. In thyroid surgery, contrast-enhanced ultrasound patterns allow tumours to be distinguished in terms of entity even before resection [14,15,16].

The objective of this prospective, monocentric clinical feasibility study was to investigate the technical feasibility and performance of Io-CEUS in minimally invasive thoracic surgery using VATS for the detection, visualization, and characterization of SPNs subsequently before surgical resection.

## 2. Materials and Methods

### 2.1. Study Design

This clinical feasibility study was a prospective, single-centre study that was carried out at a university centre for thoracic surgery, and which was conducted in close cooperation with the Institute of Radiology and the Interdisciplinary Ultrasound Center. From October 2021 to March 2023, n = 30 patients were included in whom elective, minimally invasive (VATS) lung resection was indicated due to SPNs. All patients underwent preoperative staging by contrast-enhanced computed tomography (CT) of the thorax and FDG-PET-CT if indicated. The indication for surgery was made during an interdisciplinary tumour conference. Lung surgery was performed both for histological confirmation of SPNs and, in the case of preoperative (biopsy) or intraoperative (frozen section examination) evidence of lung carcinoma, to perform an anatomic lung resection. All patients were informed preoperatively about the additional Io-CEUS and gave their written informed consent. The study received ethical approval by the local ethics committee (reference: 21-2301-101). There were no disadvantages for the patients because the intraoperative procedure, including the extent of lung resection and also the postoperative course, were not influenced by the Io-CEUS.

### 2.2. Study Hypothesis

The use of Io-CEUS in minimally invasive thoracic surgery (VATS) allows for the detection of unclear SPNs and, at the same time, visualization through the use of power Doppler and characterization by contrast kinetics. The primary endpoint was technical feasibility and performance during VATS. Secondary endpoints were defined as the detection of SPNs and, more importantly, the ultrasound-based characterization of the different pulmonary tumour entities.

### 2.3. Intraoperative Approach for Io-CEUS

All intraoperative ultrasound examinations were performed by an experienced radiologist with more than 10 years of experience (DEGUM level III), who worked in tandem with the operating thoracic surgeons under sterile conditions. Manual probe guidance was performed by the lead operating surgeon. All examinations were performed on comparable high-performance ultrasound equipment (LOGIQ E9 and LOGIQ E10, GE). The VATS was performed under general anaesthesia with single-lung ventilation using a double-lumen tube. In principle, a minimally invasive approach (VATS) was chosen that used a utility incision, where the incision was approximately 4–5 cm in the 4th or 5th intercostal space of the midaxillary line without spreading the rips. A T-probe (6–9 Mhz—L3-9i-D) was inserted via this access point. The tumour area was located via the direct application of the T-probe and visualized by means of a B-scan. Approximately 2.4–5 mL of sulphur hexafluoride microbubbles (SonoVue^®^, Bracco, Milan, Italy) was applied via bolus injections, which was followed by 10–20 mL of NaCl fed into a central venous catheter by the anaesthesiologist. Continuous digital cine loops of up to 60 s were recorded. The B-scan was optimized using frequency-dependent depth adaptation. Tissue harmonic imaging and image smoothing with SRI (spectral radiation imaging) were used. Further optimization was achieved by scanning from several angles at the same time (cross-beam technique). Colour-coded Doppler sonography (CCDS) was performed with a very low flux adaptation (pHF/scale 1000 KHz—wall filter 50). Gain was achieved with a flow that was adapted to be a low venous flow, with velocities <10 cm/s. The sweep technique was used to determine the tumour area. Time to peak (TTP) and area under the curve (AUC) were used as relative measures of the gating velocity and for estimation of dynamic contrast volumes.

## 3. Results

### 3.1. Study Population

Io-CEUS was successfully performed in a total of n = 30 patients (female, n = 13; mean age, 63 ± 8.6 years) during VATS (Table 1).

On the preoperative chest CT scan, SPNs had a mean diameter of 1.95 cm ± 1.1 SD and were located 2.02 cm ± 1.97 SD below the lung surface (visceral pleura). Indications for surgery were diagnostic for the histological confirmation of unclear SPNs (n = 21) by atypical resection, but also primary anatomic lung resection in cases of lung carcinoma detected preoperatively by biopsy (segmentectomy, n = 2; lobectomy, n = 6). In one case, segmentectomy was performed in the presence of metastasis as part of an individualized treatment approach. After an intraoperative frozen section examination, lung carcinoma was confirmed in eight patients, and so anatomic lung resection (lobectomy, n = 5; segmental resection, n = 2) was performed. In one patient, only enucleation was performed for a typical carcinoid (size, 0.4 cm). In four cases, segmental resection was performed due to the size and location of the SPN when metastasis was detected (Table 2).

### 3.2. Performance of Io-CEUS

In all cases, the SPNs could be detected and visualized with the ultrasound probe (100%). No patient was noted to have a contrast intolerance reaction during Io-CEUS. Based on the contrast distribution pattern (peripheral and/or central enhancement) as well as the onset and duration until maximal contrast uptake, the characterization of the SPNs was performed.

### 3.3. Lung Carcinomas

A total of n = 16 primary lung carcinomas were visualized by Io-CEUS (Table 3). The lung carcinomas were located in all lobes of the lung. These were measured via a preoperative CT scan, and had a mean size of 1.98 cm ± 1.03 SD and a mean distance to the visceral pleura of 1.99 cm ± 1.77 SD. Primary lung carcinomas had jagged margins in B-mode and were predominantly inhomogeneous in echogenicity. Evidence of necrotic areas was present in some tumours, especially in larger tumours (>3 cm). After neoadjuvant chemotherapy (n = 1), large necrotic areas could be delineated. CCDS or power Doppler showed both peripheral and central macrovascularization in the primary lung carcinomas. On CEUS, the primary lung carcinomas showed an early peripheral and central wash in, with the mean at t = 8.8 s ± 3 SD and a maximal distribution mean after t = 20 s, excluding the necrotic areas if present.

### 3.4. Lung Metastases

Lung metastases (n = 11) were located in all lobes of the lung (Table 3). On the CT scan, they had a mean size of 1.76 cm ± 0.84 SD and a distance of 2.24 cm ± 2.03 SD from the lung surface. The ultrasound pattern of the lung metastases showed a partly smooth and partly blurred border in B-mode. Echogenicity was mostly hypoechogenic, but sometimes echo-inhomogeneous. In CCS or power Doppler, peripheral macrovascularization could be detected in nine cases, and in three of these patients it was even detected ubiquitously. Two cases showed no macrovascularization. On the CEUS, the lung metastases mostly presented with marginal contrast enhancement. In some cases, early contrast enhancement was evident (t = 6 s), but in others it was late (t = 16 s). In most cases, enhancement was confined to the margin of the tumour. In three cases, a subtotal spread of the contrast agent occurred.

### 3.5. Benign Nodule

Three benign SPNs (right upper lobe, n = 2; right lower lobe, n = 1) with a mean size of 1.13 cm were examined. The mean distance to the lung surface determined via imaging was approximately 1.06 cm. The benign SPNs had a partly sharp but also partly blurred border, and showed hypoechogenic to echo-inhomogeneous ultrasound patterns. CCDS and power Doppler demonstrated marginal macrovascularization in two cases and ubiquitous macrovascularization in one case. CEUS showed early (t = 4 s) and diffuse contrast enhancement with a subtotal spread in two cases. In another case, late (t = 29 s) and irregular enhancements were detected.

### 3.6. Case 1

A 59-year-old patient had an unclear SPN (approx. 1.9 cm in size) in the left lower lobe, which was in direct contact to the visceral pleura (Figure 1). In the interdisciplinary tumour board, the indication was given for surgical histological confirmation and, in the presence of bronchial carcinoma, anatomic resection. The B-mode showed a blurred tumour with an inhomogeneous ultrasound pattern (Figure 2). In CCDS, several vessels could be delineated within the tumour. When Io-CEUS was performed, early marginal and central contrast enhancement occurred (t = 5 s). Rapidly (t = 8 s), the enhancement extended throughout the tumour. The frozen section confirmed the presence of an adenocarcinoma, and thus a lobectomy of the left lower lobe of the lung was performed. The final histology confirmed that it was a pulmonary adenocarcinoma with an extension of 2.3 cm.

### 3.7. Case 2

A 71-year-old female patient had bipulmonary SPNs that needed histologic confirmation. Based on the CT scan, an SPN in the left upper lobe of the lung was selected for resection (Figure 3). In B-mode, the tumour was visualized with sharp margins and a predominantly hypoechogenic internal structure. In the marginal area, the tumour showed hyperechogenicity. Native vascularization could be demonstrated in the marginal area of the tumour. Io-CEUS measured an enhancement onset from t = 12 s, which was limited to the tumour margin (Figure 4). There was no contrast enhancement centrally. The final histology confirmed a lung metastasis of a leiomyosarcoma with a diameter of 0.9 cm.

## 4. Discussion

This feasibility study proved the technical feasibility of Io-CEUS in minimally invasive thoracic surgery via VATS. Furthermore, the ability of Io-CEUS using power Doppler as well as contrast agents seems to already provide encouraging results regarding the intraoperative assessment of pulmonary tumours of various histologies. This represents a new and innovative intraoperative approach in thoracic surgery, which will allow for collecting unique data worldwide via Io-CEUS for the specific differentiation of particularly malignant pulmonary tumours.

Our first results and practical experience confirm that pulmonary tumours of different sizes (0.6 cm–4.6 cm) and a maximum distance of 6.4 cm to the lung surface can be reliably found intraoperatively by B-mode. This has also been demonstrated in other studies compared to results obtained via manual palpation [3]. Previous studies primarily investigated the value of intrathoracic ultrasound in terms of its ability to detect pulmonary nodules and compared the results with intraoperative palpation [3,5,17]. In particular, Khereba et al. demonstrated the high utility of intraoperative ultrasound for pulmonary nodule detection in minimally invasive thoracic surgery [6]. This reduced the rate of conversions from VATS to conventional open thoracotomy, which has a limited manual palpation capability as a minimally invasive approach. Increasingly, robot-assisted procedures are being adopted in thoracic surgery [18,19,20,21].Currently, the lack of palpable control when using this robotic system is a major disadvantage, as surgeons are unable to locate lesions with their finger or a device as is possible with VATS.

The insertion of the T-probe into the chest via a minimally invasive access point of about 4 cm in length was technically possible. However, we must note that the intraoperative and intrathoracic handling of the T-probe via such a small approach point is technically difficult, and so an endoscopic ultrasound probe with the necessary functions should be used here in the future. However, it would then also be a prerequisite that this endoscopic probe has other functions which are absolutely necessary for the specific description of the tumour. In our study, we increasingly focused on the visualization as well as characterization of pulmonary tumours, using B-mode and CCDS or power Doppler to visualize native vascularization and CEUS to visualize microvascularization. B-mode alone did not differentiate tumours with respect to malignancy. However, our data provide preliminary evidence of differential vascularization as well as contrast kinetics depending on the tumour entity. Primary lung carcinomas showed a predominantly early central and concurrent peripheral onset contrast enhancement with a subtotal filling of the entire tumour over time. Of particular interest, the different tumour entities of primary lung carcinomas could not be distinguished based on their contrast agent behaviour. In contrast, clear differences emerged in the lung metastases with enhancement confined to the tumour margin and were limited to this over time. However, ubiquitous diffuse contrast enhancement occurred in three cases. The cases were metastases from a malt lymphoma, a liposarcoma, and a urothelial carcinoma. Benign lesions were highly variable as determined with B-mode and CCDS, or power Doppler and CEUS. Ultrasound patterns ranged from hypoechogenic to echo-inhomogenic. Native vascularization was seen in addition to microvascularization, from the marginal to the diffuse subtotal. With a total of only three cases examined, we do not believe that any conclusions can be drawn regarding specific characterization.

Meanwhile, the utility of intraoperative ultrasound during liver tumour surgery for the detection, characterization, and visualization of the extent of malignancy-specific liver tumours, as well as for marking resection and ablation margins and monitoring tumour ablation, has been well studied [9,11,22,23,24,25,26,27]. In addition to the use of CCDS and power Doppler, Io-CEUS can contribute valuable information regarding liver perfusion in general, as well as that related to vascularization and thus the entity determination of liver malignancies. Furthermore, the surgical approach can be significantly influenced by the use of Io-CEUS [12,23,25,28,29,30].

However, the blood supply to the liver differs from that to the lungs. Here, an arterial, portal venous, and late venous phase can be distinguished. Based on wash-in and wash-out kinetics, the liver lesion can be characterized, with an arterial hypervascularized or irregularly vascularized lesion with wash-out in the portal venous and late phase being classified as malignant. Increasing the wash-in to the portal venous and late phase is considered a criterion of benignity [13,31]. The lung, on the other hand, presents an entirely different perfusion situation with the vasa privata and publica. Here, contrast enhancement depends on the perfusion of the tumour by the lung’s own vessels or by the pulmonary arteries. From which vascular system the tumour is fed is currently unclear. A wash-out has been observed only in isolated cases. A possible hypothesis would be that an early and rapid contrast enhancement, for example from t = 6 s, is observed in tumours, which has a direct connection to the pulmonary arterial supply, whereas a later enhancement is supplied by the bronchial arteries. Tumour perfusion turns out to be complex and multimodal. This should be explored in further studies.

The first successful application of an endoscopic ultrasound probe in VATS by our research group confirmed the technical feasibility and the potential for further investigations (Figure 5) [4,17,32]. From this perspective, Io-CEUS may become a very helpful and indispensable tool, especially in VATS, but also in robotic-assisted thoracic surgery (RATS) [33]. Since manual palpation is often not possible in these minimally invasive procedures, Io-CEUS could detect even small SPNs and at the same time visualize as well as characterize them using various techniques, almost in the sense of a “live histology”. These intraoperative results would have a direct impact on the resection procedure. Our research group is continuously working on the intraoperative application of Io-CEUS as well as on further innovative methods of high performance sonography, which includes, e.g., elastography, 3D imaging, and fusions to CT imaging.

## 5. Conclusions

In conclusion, Io-CEUS using a T-probe is safely applicable in minimally invasive thoracic surgery (VATS) before lung resection. The differentiation between primary lung carcinomas and metastases directly before surgical resection seems possible with the help of high-performance sonography, including CCDS, power Doppler, and CEUS under the assessment of contrast kinetics. Consequently, in the future, unclear pulmonary tumours can be not only detected, but also characterized with Io-CEUS and can provide intraoperative clues regarding the entity. Io-CEUS thus has the potential to influence the extent of resection even before the frozen section result and could revolutionize thoracic surgical resections that are increasingly performed via VATS or RATS. For a better differentiation, elastography should be applied additionally.

## Figures and Tables

**Figure 1 cancers-15-03854-f001:**
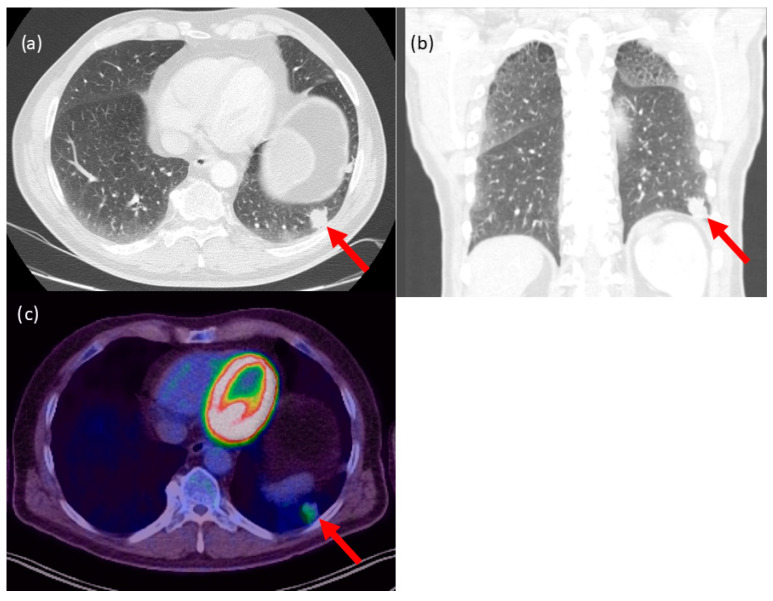
Case with lung carcinoma in the left lower lobe. Contrast-enhanced CT scan with a slice thickness of 0.5 mm, (**a**) axial and (**b**) coronary view. (**c**) FDG-PET-CT scan of the tumour with increased metabolism. The red arrow marks the tumour.

**Figure 2 cancers-15-03854-f002:**
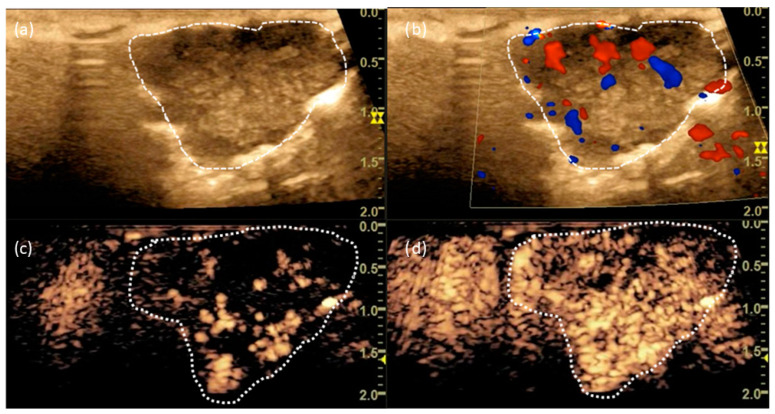
(**a**) B-mode and (**b**) CCDS showing macrovascularization. (**c**) CEUS at t = 6 s and (**d**) t = 8 s showing marginal and central enhancement. The dashed line marks the border of the tumour.

**Figure 3 cancers-15-03854-f003:**
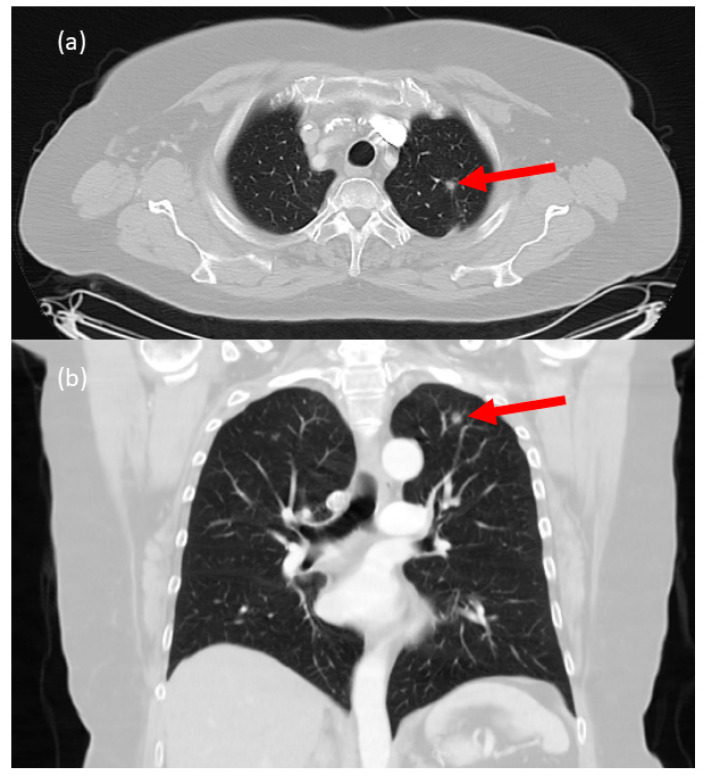
Contrast-enhanced CT scan of an SPN in the left upper lobe of the lung (**a**) axial and (**b**) coronal plane. Red arrow marks the tumour.

**Figure 4 cancers-15-03854-f004:**
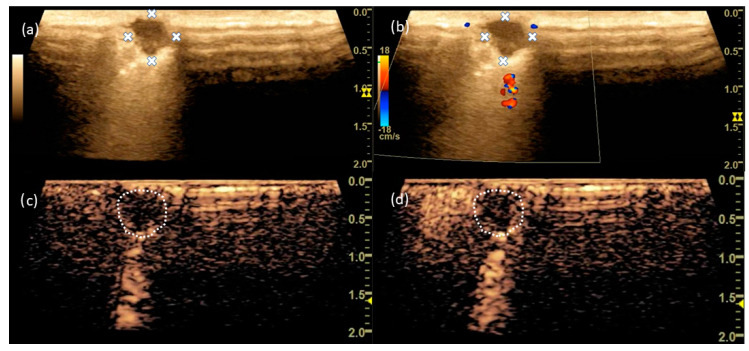
In B-mode, the SPN shows a clean margin (**a**) with a hyperechogenic edge region and single-vessel gating in the edge region (**b**) in CCDS. CEUS shows enhancement restricted to the border region (**c**) at t = 15 s and (**d**) at t = 27 s. The dashed line and the x marks the border of the tumour.

**Figure 5 cancers-15-03854-f005:**
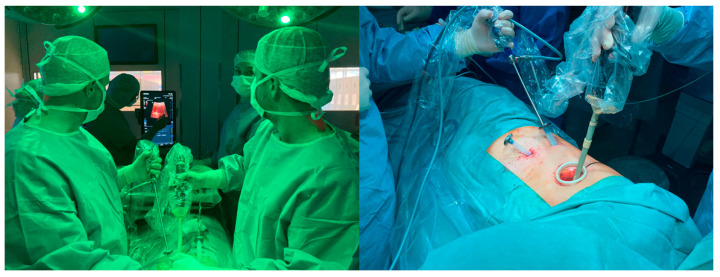
Intraoperative intrathoracic application of Lap 13-4cs laparoscopic probe on TE9 (Mindray, Shenzhen, China) via minithoracotomy.

**Table 1 cancers-15-03854-t001:** Study population.

	Total n = 30 (100%)
Gender (female: n; %)	14 (47%)
Age (mean ± SD; years)	64 ± 8.8
Side of surgery (n; %)	
right	15 (50)
left	15 (50)
Localization of investigated suspicious SPNs (n; %)	
upper lobe	18 (60)
middle lobe	3 (10)
lower lobe	9 (30)
Diameter in CT-scan of suspicious SPNs (mean ± SD) (cm)	1.95 ± 1.1
Distance to lung surface (mean ± SD) (cm)	2.02 ± 1.97

**Table 2 cancers-15-03854-t002:** Intraoperative data.

	Total n = 30 (100%)
Indication for surgery (n; %)	
diagnostic	21 (70)
therapeutic	9 (30)
Frozen section (n; %)	n = 21 (70)
primary lung carcinoma	8 (38)
metastasis	8 (38)
benign lesion	3 (14)
no frozen section	2 (10)
Lung resection (n; %)	
enucleation	1 (3)
wedge resection	9 (30)
segmentectomy	9 (30)
lobectomy	11 (37)
Duration of surgery (mean ± SD; min)	146 ± 58
Detection with Io-US (n; %)	30 (100)
Performance of Io-CEUS (n; %)	30 (100)
B-mode	30 (100)
Power Doppler	30 (100)
CEUS	30 (100)

**Table 3 cancers-15-03854-t003:** Histological results.

	Total n = 30 (100%)
**Lung carcinoma (n; %)**	**16 (53)**
Adenocarcinoma	10 (33)
Typical carcinoid	4 (13)
Squamous cell carcinoma	2 (7)
**Pulmonary metastasis (n; %)**	**11 (37)**
Squamous cell carcinoma	2 (7)
Renal cell carcinoma	1 (3)
Cholangiocellular carcinoma	1 (3)
Sarcoma	1 (3)
Colon carcinoma	1 (3)
Lymphoma	1 (3)
Pancreatic carcinoma	1 (3)
Leiomyosarcoma	1 (3)
Liposarcoma	1 (3)
Malignant melanoma	1 (3)
**Benign lesion (n; %)**	**3 (10)**
Granuloma inflammatory	1 (3)
Granuloma rheumatoid	1 (3)
cryptogenic pneumonia	1 (3)
Tumour size (mean ± SD; cm)	2.16 ± 1.12
Tumour size (mean ± SD; cm)	
Lung carcinoma	2.80 ± 1.03
Pulmonary metastasis	1.76 ± 0.84
Benign lesion	1.13 ± 0.32

## Data Availability

Data are available upon request due to privacy and ethical restrictions.

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
