# Peer review of "Intraoperative Contrast-Enhanced Ultrasonography (Io-CEUS) in Minimally Invasive Thoracic Surgery for Characterization of Pulmonary Tumours: A Clinical Feasibility Study"

_cancers, 2023, doi:10.3390/cancers15153854_

Round 1

Reviewer 1 Report

Thanks for inviting me to review this pioneer study.  The authors were trying to show the potential benefits in differential diagnosis especially using of intraoperative contrast enhanced ultrasound (Io-CEUS), in addition to the impact of ultrasound in detecting pulmonary nodule which has been demonstrated in other studies. I think this effort should be encouraging when you think about that in the future your diagnostic algorithm for SPNs may be totally differently as what we currently have. And it may be possible that you don’t proceed patients with SPNs for CT scan or FDG PET scan in the future which is considered as standard tools now.

However, the authors should convince readers more potential benefits of this novel imaging modality.  

1. Due to limited cases numbers, the characters of Io-CEUS in lung metastasis or benign lesions are equivocal. Is it possible to add more cases so that we may can see a more clear US pattern in those tumor entities?

2. Do you have the data such as size of tumors, or distance to visceral pleura which gathered from intraoperative ultrasound, to show if corresponding to those gathered from pathology? I think it is meaning to show these data since it can help with determining the extent of resection if it is accurate.

3. Do you have a case presentation to show that the application of intraoperative US changed your decision at the OR?

4. Comments about case 1, I know you want to show the typical US pattens in primary lung cancer. However, readers may would like to know if biopsy or pathology has been known before the surgery in this easily accessed, peripheral nodule.

5. Can you comment if your probe and minimal thoracotomy change the incision designs in robotic surgery? I mean we don't make a 3-4cm incision during robotic surgery. 

Author Response

Dear reviewer

Thank you very much for the helpful comments.
Attached are the answers to your comments. We hope that they have been answered and processed to your satisfaction.

Kind regards
Martin Schauer

Reviewer 2 Report

Dear Editor and Authors,

Thank you for asking me to review this manuscript titled “Intraoperative contrast-enhanced ultrasonography (Io-CEUS) in minimally-invasive thoracic surgery for characterization of pulmonary tumours: a clinical feasibility study” by Dr. Schauer and colleagues from the Department of Thoracic Surgery at the University Medical Center Regensburg, in Regensburg, Germany.

In this prospective, single center feasibility study the authors utilized intraoperative contrast- enhanced ultrasonography (Io-CEUS) to locate and identify solitary pulmonary nodules (SPN) in 30 patients undergoing minimally invasive/thoracoscopic surgery. Using the Io-CEUS, SNPs were identified in all patients and where successfully resected.

I have the following commentary to make after evaluation it.

Comments:

1.    What does “increasingly upcoming screening examinations” mean? I understand the authors want to comment about the “over the years increased prevalence of lung cancer screening” but they should rephrase it better.

2.    I am not sure we need “innovative procedures” to find SNPs, I would rather have reliable and affordable procedures!!

3.    What is “unclear dignity”, it does not make sense!!

4.    I would not call 5 studies “a few”!! I think there is quite a significant body of evidence of US guided localization of SNPs!! I would rephrase this whole section in the introduction.

5.    What are “desufflated lungs”? Do you mean deflated?

6.    What is “a positive ethics vote”? Do you mean ethical approval?

7.    Use lead surgeon instead of 1st surgeon!!

8.    If VATS was performed then a mini-thoracotomy was not performed (thoracotomy means use of retractor!!), use utility incision instead.

9.    Why was a central line inserted in patients undergoing VATS surgery?? The whole point of minimally invasive surgery is to eliminate unnecessary procedures that increase patient risk!! Was it needed for the injection of the contrast agent? Could this not be done via a peripheral line?

10.  What is “individualized treatment concept”? Do you mean individualized treatment approach?

11.  There is a discrepancy in the results section between the number of reported lobectomies in the text and in the table (one says 6 and the other 9!!). Same with segmentectomies (9 versus 4+2!!).

12.  What is “Srcoma” in table 3? Do you mean sarcoma?

13.  Case 1 and Case 2 are irrelevant and have nothing to offer to the article. I suggest they are eliminated!!

14.  The manuscript needs some significant language and expression editing prior to publication.

In conclusion, this is an interesting article that offers valuable information to the surgical community but needs some minor revision prior to publication. Thank you.   

Please see my comments - it needs some significant editing!

Author Response

(The authors gave the same response as above.)
